# Altered Feedback-Related Negativity in Mild Cognitive Impairment

**DOI:** 10.3390/brainsci13020203

**Published:** 2023-01-25

**Authors:** Satoshi Abe, Keiichi Onoda, Masahiro Takamura, Eri Nitta, Atsushi Nagai, Shuhei Yamaguchi

**Affiliations:** 1Department of Neurology, Faculty of Medicine, Shimane University, Izumo, Shimane 693-8501, Japan; 2Department of Psychology, Otemon Gakuin University, Ibaraki, Osaka 567-8502, Japan; 3Laboratory Medicine, Shimane University Hospital, Izumo, Shimane 693-8501, Japan; 4Department of Neurology, Shimane Prefectural Central Hospital, Izumo, Shimane 693-8555, Japan

**Keywords:** feedback-related negativity, event-related brain potentials, mild cognitive impairment

## Abstract

Introduction: Feedback-related negativity (FRN) is electrical brain activity related to the function of monitoring behavior and its outcome. FRN is generated by negative feedback input, such as punishment or monetary loss, and its potential is distributed maximally over the frontal-central part of the skull. Our previous study demonstrated that FRN latency was delayed and that the amplitude was increased in patients with mild Alzheimer’s disease (AD). As mild cognitive impairment (MCI) is considered to be a prodromal stage of AD, we speculated that FRN would also be altered in MCI, as in AD. The aim of this study is to examine whether MCI patients showed changes in FRN during a gambling task. Methods: Thirteen MCI patients and thirteen age-matched healthy elderly individuals participated in a simple gambling task and underwent neuro-psychological assessments. The participants were asked to choose one out of two options and randomly received positive or negative feedback to their response. An EEG was recorded during the task, and FRN was obtained by subtracting the positive feedback-related activity from the negative feedback-related activity. Results: The reaction time to probe stimuli was comparable in the two groups. The group comparisons revealed that the FRN amplitude was significantly larger for the MCI group than for the healthy elderly (*F*(1,24) = 6.4, η_p_^2^ = 0.22, *p* = 0.019), but there was no group difference in the FRN latency. The FRN amplitude at the frontocentral electrode positively correlated with the mini-mental state examination score (*Spearman’s rho_partial_* = 0.41, *p* = 0.043). The finding of increased FRN amplitude in MCI was consistent with the previous finding in AD. Conclusion: Our findings indicate that monitoring dysfunction might also be involved in the prodromal stage of dementia.

## 1. Introduction

Patients with mild cognitive impairment (MCI) complain of memory impairment and show slight cognitive deficits in formal neuropsychological assessments but do not fulfill the criteria for a diagnosis of dementia. Approximately 50% of MCI patients develop dementia in 5 years [1], and 10–15% of MCI cases convert to Alzheimer’s disease (AD) annually, as opposed to the 1–2% conversion rate among the healthy elderly [2]. Pathological processes, i.e., the deposition of beta-amyloid, have already occurred in the brain of MCI patients, although the cognitive deterioration remains modest [3]. Cognitive impairments in MCI patients are seen in at least one functional domain, such as memory, language, executive function, or visuospatial ability [4,5,6].

Although intact daily functioning is one of the defining criteria for MCI, a number of recent investigations have reported that MCI patients perform poorly compared to their healthy aging peers in terms of some complex activities in daily life, such as bank statement management and bill payment [7] and in arithmetic tasks when there is a high load on executive functions [8]. Executive functions are mental processes that enable us to control and guide goal-oriented behaviors effectively. The mental processes include planning, working memory, inhibition, mental flexibility, initiation, and the monitoring of action [9], which are essential processes for achieving a goal. The monitoring process includes appropriate assessment of feedback information regarding environmental changes by one’s own action and is critical for survival and social adaptation. 

In the past two decades, brain processes related to feedback information have been a research focus. Feedback-related negativity (FRN) is generated by external feedback, typically in a gambling task [10,11,12] or a time-production task [13,14,15], with a latency range of 200–300 ms after the feedback signal [10,13,16,17,18]. The source of FRN generation is speculated to be in the anterior cingulate cortex [17,19,20,21], and its topography is distributed maximally over the frontal-central scalp area. A number of studies have reported that FRN alters with age, similarly to other ERPs. FRN amplitude is reduced in the elderly compared to young people [14,22,23,24], and its latency is prolonged in the elderly [24], although studies do not fully agree on the effects of aging on FRN [25]. Furthermore, changes in FRN in cognitively impaired subjects have not been fully examined. In our previous study [26], FRN latency was delayed, but its amplitude paradoxically increased in patients with mild AD compared to healthy elderly. Based on this finding in AD, the current study aimed to examine the changes of FRN in individuals with MCI. As MCI is considered as a prodromal stage of AD, we hypothesized that FRN may show alterations in MCI similar to AD.

## 2. Materials and Methods

### 2.1. Subjects

We examined 13 amnestic MCI patients (MCI group; 6 males and 7 females, mean age 76.2 ± 2.4 (SD) years) and 13 age-matched healthy elderly volunteers (HC group; 6 males and 7 females, mean age 74.1 ± 3.4 years). All MCI patients consulted our hospital complaining of forgetfulness and were diagnosed according to the revised Petersen’s criteria [27]. All participants were evaluated using the clinical dementia rating (CDR) and mini-mental state examination (MMSE) [28]. The diagnosis of MCI was made for participants in whom the overall score of the CDR was 0.5, the memory score on the CDR was 0.5 or 1, and the MMSE score was between 24 and 30 points. In addition, subjective memory impairment was self-reported or reported by informants without evidence of functional decline, associated with or without impairment in other non-memory cognitive domains. The criteria for the HC individuals was a CDR of 0 and MMSE score above 27 points. All participants had normal or corrected-to-normal vision and had no history of other neurological and psychological diseases. All participants underwent conventional MRI examination, and no participant demonstrated ischemic or hemorrhagic cerebrovascular lesions or marked white matter lesions. This research was approved by the ethics committee of Shimane University. All subjects agreed to participate in this study by providing written informed consent.

### 2.2. Neuropsychological Assessment

In addition to the MMSE and CDR assessments, we performed the following neuropsychological tests: frontal assessment battery (FAB) [29], self-rating depression scale (SDS) [30], and apathy scale (AS) [31].

### 2.3. Task

Participants took part in a simple gambling task. Each participant sat in front of a computer screen, approximately 1.5 m away from the screen, in an electrically shielded, dimly lit, and sound-attenuated room. At the start of the trial, green and purple squares were displayed on the left and right side of the screen randomly, and the participants selected 1 of the 2 squares by pressing a corresponding button (left or right). The squares were shown until the button press response. Then, positive or negative feedback was randomly presented in the screen center at 1500 ms after the button press for a period of 1000 ms. A JPY 100 coin (approximately USD 1) was the positive feedback, and a red cross over the coin was the negative feedback. These corresponded to a win and lose condition, respectively, and the probability of each condition was equal, irrespective of participants’ choice. Participants were asked to maximize the amount of virtual monetary reward, although they were not informed that the probability of winning was 50%. They practiced for 10 trials before starting the experiment. They completed 120 trials (two blocks of 60 trials each) in the experiment, with an inter-trial interval of 2–3 s. An electroencephalogram (EEG) was recorded during task performance.

### 2.4. EEG Recording and Analysis

EEG was recorded with 21 channels based on the international 10/20-system: Fp1, Fpz, Fp2, F7, F3, Fz, F4, F8, T3, C3, Cz, C4, T4, T5, P3, Pz, P4, T6, O1, Oz, and O2, and referenced to the linked mastoids. A vertical electro-oculogram (EOG) was recorded from the supra- and infra-orbit of bilateral eyes, and a horizontal EOG was recorded from the external canthi of each eye in order to detect eye movements. Electrode impedance was maintained below 5 kΩ. EEG signals were recorded continuously with bandpass filtering at 0.01–250 Hz and were amplified using the BrainAmp amplifier (Brain Products, Munich, Germany) with the appropriate software. The data were collected at a sampling frequency of 500 Hz. Reaction time (RT) to the choice stimulus was recorded simultaneously using EEG. 

### 2.5. Event-Related Potentials

EEG data were analyzed off-line using the Brain Vision Analyzer 2 software (Brain Products, Munich, Germany). An independent component analysis (ICA) was performed on single-subject EEG data in order to correct blink artifacts. All segments exceeding a ±100 µV threshold were rejected as artifacts. EEGs were averaged over 1000 ms and time-locked to the onset of feedback stimuli, including 200 ms of the pre-stimulus baseline. Segmented EEG for negative and positive feedback stimuli were averaged separately. FRN was obtained by subtracting the ERP to the positive feedback from that of the negative feedback. FRN was measured at the most negative peak in the time-window from 250 to 400 ms after feedback presentation at the midline electrode sites (Fz, Cz, and Pz). The amplitude of FRN was assigned as the absolute peak value of the wave relative to the prestimulus baseline. 

### 2.6. Statistical Analysis

We performed t-tests for demographic (except for using a χ^2^ test for gender), neuropsychological, and RT data to compare the HC and MCI groups. Repeated-measure ANCOVA was performed for the amplitudes and latency of ERP components, using the group as the between-subject variable and the channel as the within-subject variable with age as a covariate. The threshold for statistical significance was set to a *p* value less than 0.05. Partial correlation analyses were also conducted to examine relationships between the ERP components and the neuropsychological data. The analyses were conducted using non-parametric method (Spearman’s correlation), with adjustment for age. SPSS (ver. 23; SPSS Inc., Chicago, IL, USA) was used for the statistical analysis. The Bonferroni method was used for correction of multiple comparisons.

## 3. Results

### 3.1. Neuropsychological and Behavioral Data

The demographic characteristics of the participants are listed in Table 1. There were no significant differences in age and the gender ratio between the MCI and HC groups. The MMSE and FAB scores were significantly lower for the MCI group than for the HC group (*ts*(24) > 4.0, *Cohen’s ds* > 1.5, *ps* < 0.001). The apathy score was significantly higher for the MCI group compared to the HC group (*t*(24) = 2.5, *Cohen’s d* = 1.0, *p* = 0.021), while the depression scale was comparable between the two groups. There was no significant difference in the RT to the choice stimuli between the MCI and HC groups. 

### 3.2. ERP Waveforms

Figure 1 demonstrates the grand-average waveforms at Fz, Cz, and Pz to the feedback stimuli for the win and lose conditions. N2 was evoked at the time-window of 200–400 ms, and P3 was at 300–500 ms in both the win and lose conditions. The ERP waveform for the lose condition was negatively shifted within the time-window of 200–300 ms compared to the win condition in both the MCI and HC groups. The negative shift for the lose condition was larger in the MCI group than in the HC group across three channels. The statistical group comparison was performed on the subtraction waveform, i.e., FRN.

### 3.3. FRN

Figure 2 shows the subtraction waveforms for the win trials from the loss trials. FRN was observed as a negative deflection in the 200–300 ms time-window. We performed repeated-measure ANCOVA for the peak amplitude and latency of the FRN. The main effect of the group was significant for the FRN amplitude (*F*(1,24) = 6.4, η_p_^2^ = 0.22, *p* = 0.019, observed power = 0.7), with a larger negative peak for the MCI group than for the HC group (mean amplitude of FRN across three channels: −5.83 ± 2.17 µV for MCI and −2.69 ± 2.47 µV for HC) (Figure 3). Meanwhile, there were no significant differences in the FRN peak latency between the two groups. 

### 3.4. Neuropsychological Assessment and FRN

Partial correlation analyses were performed between the neuropsychological data and FRN across all subjects, with adjustment for age. The FRN amplitude at Fz was slightly correlated with the MMSE score (*Spearman’s rho_partial_* = 0.41, uncorrected *p* = 0.043, Figure 4), indicating that subjects with a lower MMSE score showed a larger negative amplitude of FRN. However, when we added the group as another covariate, the correlation was diminished. The FAB, SDS, and AS scores did not show any significant correlations with any measures for FRN (Table 2).

## 4. Discussion

This study demonstrated that FRN amplitude was significantly increased in amnesic MCI presumed to have AD pathology [27]; this has not been reported previously. The original hypothesis of this study was that FRN would already be augmented in MCI patients because mild AD patients demonstrated increased FRN amplitude compared to elderly healthy controls in our previous study [26]. This suggested that the mechanism underlying the change in FRN amplitude along with cognitive impairment may be linked to the neurodegenerative process, rather than simple aging per se. FRN is implicated in the monitoring of one’s own behavior, which might be distinctively affected by the process of neural degeneration compared to other cognitive functions, such as memory function. 

The most plausible explanation for the current finding is that the phase of increased slow wave was reset by the feedback stimulus. The theta waves in the frontal area were generated in the dorsal anterior cingulate cortex [32]. FRN is generated by the phase-locked theta waves originating from the region [33]. The EEG studies on AD have reported an increase in the slower frequency band (including theta) and a decrease in the faster frequency bands (alpha and beta); studies on MCI have found that these patients share similar EEG characteristics as AD [34]. Musaeus et al. [34] discussed that the increases in the relative theta power could be a sign of the underlying network dysfunction in patients with cognitive deficit.

Another explanation for the current finding is that a compensatory mechanism against cognitive decline may contribute to the increase in FRN amplitude. Functional MRI (fMRI) studies have demonstrated that some parts of the brain are more active in patients with MCI than in healthy elderly subjects. Bakker et al. reported that individuals with MCI showed increased fMRI BOLD activation in the medial temporal lobe during a memory task [35]. Another neuropathological study also demonstrated that certain cholinergic pathways in the superior frontal cortex were selectively upregulated, in addition to those in the hippocampus, in MCI patients [36]. These findings suggest that the neural reserve during early cognitive decline may enable individuals with MCI to increase neural activity in order to compensate for the damage in some brain regions. In addition, the frontal cortex, which is a brain region responsible for monitoring function, is affected later during dementia progression, as evidenced in a pathological study [37]. These neural compensatory responses might explain the increased FRN amplitude in MCI patients. The positive correlation between FRN amplitude and MMSE is in line with the above interpretation. However, the correlation analysis in the present study was exploratory and requires future validation using rigorous statistical criteria.

The current task paradigm was designed to elicit FRN by providing feedback on negative information. While a gambling task employed for an FRN study assesses the process of decision-making, the emotional response during the task, i.e., positive or negative emotion elicited by reward gain or loss, might be overlapped in the FRN amplitude. Negative information could generate greater brain responses than positive information. One possible explanation for the augmented amplitude of FRN is that the amplitude could be increased given that the response to positive feedback is reduced relative to negative stimuli in the patient group. Reward positivity has been reported to increase, as a function of reward, contingencies only in the young and not in the elderly [38]. The ERP changes to reward contingencies should be systematically investigated in association with the dementia processes in future. The alteration of FRN by differences in responses to positive and negative feedback has been reported in other experimental situations. AD patients are reported to show even enhanced skin conduction responses to emotional stimuli compared to healthy controls [39]. Thus, the implicit processing of emotional information, especially information with a negative valence, may be relatively resistant to degenerative processes in the prodromal stage of dementia [40].

In contrast to the amplitude change, the latency of FRN was not delayed in individuals with MCI. Our previous study demonstrated a prolonged latency of FRN in patients with mild AD compared to a younger healthy control group but not to a healthy older group. The latencies for several ERP components, such as N2 or P3, were not affected in individuals with MCI compared to those with AD [41]. Thus, ERP latency seems to have less sensitivity for detecting early changes in the speed of information processing in the pre-stage of dementia, especially when age effect is controlled. Our findings suggest that alterations in neural activity to negative stimuli (i.e., amplitude) may precede the decrease in processing speed in MCI.

There are several limitations in this study. First, the number of participants was small. We have conducted a post-hoc power analysis to confirm the validity of the sample size of 13 cases for each group. Although the observed power of the main result has reached 0.7, the value is not necessarily high enough. The finding should be evaluated further in studies comprising a large number of MCI patients. Secondly, the underlying pathology of MCI was uncertain, although we recruited amnestic MCI patients by means of the conventional criteria [27]. Our target was subjects with MCI due to early AD, whose diagnosis usually requires additional information from PIB-PET, FDG-PET, and cerebrospinal fluid biomarkers to ensure an underlying AD pathology. We may need to follow-up on our subjects longitudinally to confirm that their MCI was due to underlying AD.

## 5. Conclusions

Our previous study demonstrated that mild AD patients showed increased FRN amplitude, suggesting the existence of a compensatory mechanism against the decline in cognitive function. The current results indicated that MCI patients also had augmented neural activity evoked by negative feedback information, possibly reflecting an intact neural reserve mechanism against cognitive deterioration. The alteration in FRN amplitude could be a good biomarker for the early detection of dementia, and this study provided one of the cornerstone findings.

## Figures and Tables

**Figure 1 brainsci-13-00203-f001:**
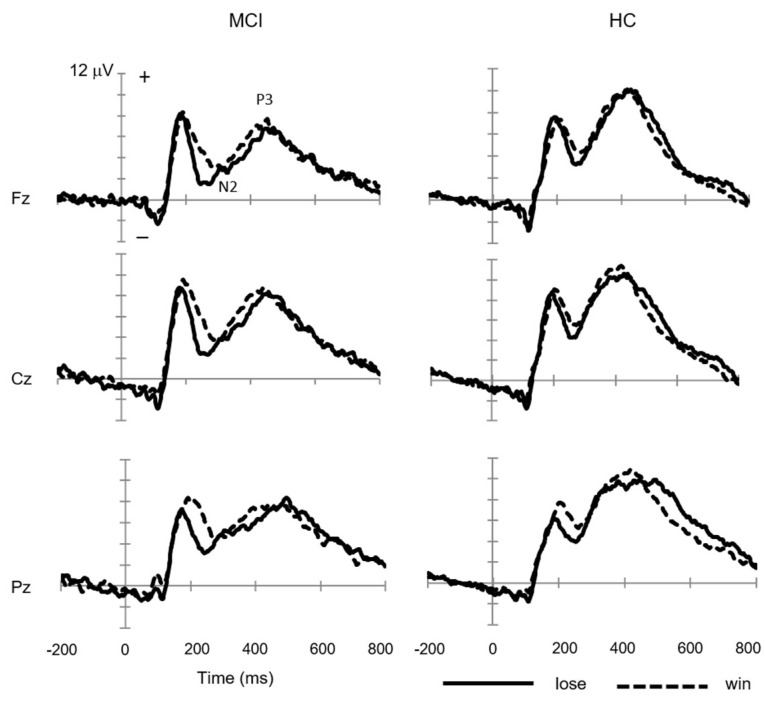
Grand-average waveforms at Fz, Cz, and Pz to feedback stimuli for win and lose trials in mild cognitive impairment (MCI, *n* = 13) and healthy control (HC, *n* = 13) groups.

**Figure 2 brainsci-13-00203-f002:**
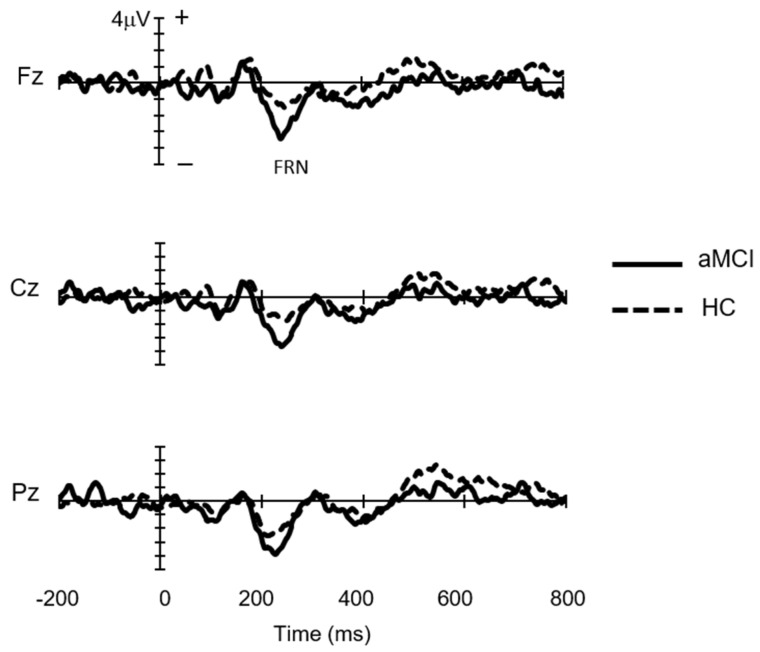
Grand-average waveforms at Fz, Cz, and Pz to positive feedback stimuli subtracted from negative feedback stimuli in mild cognitive impairment (MCI, *n* = 13) and healthy control (HC, *n* = 13) groups. FRN: feedback-related negativity.

**Figure 3 brainsci-13-00203-f003:**
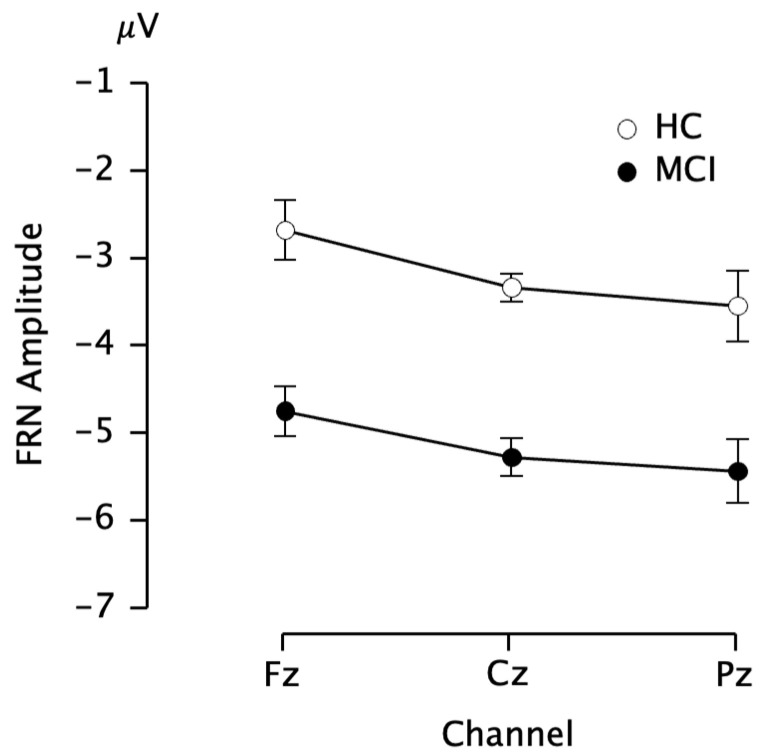
Comparisons of FRN amplitude between mild cognitive impairment (MCI) and healthy control (HC) groups. Values are mean ± S.E. FRN: feedback-related negativity.

**Figure 4 brainsci-13-00203-f004:**
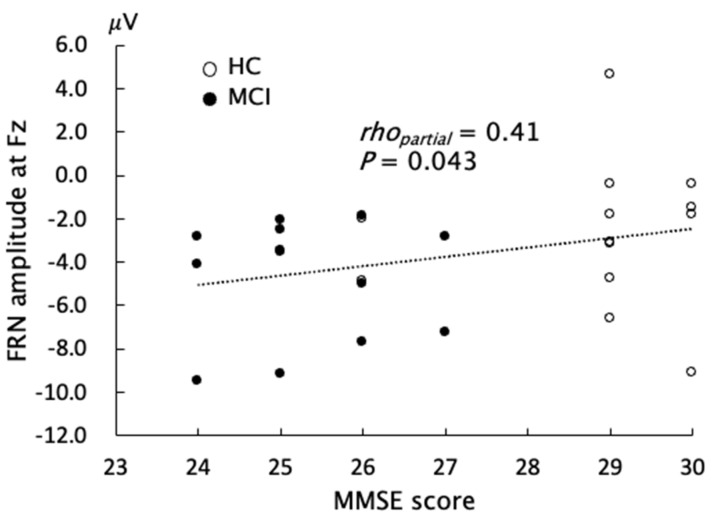
Correlations between the mini-mental state examination (MMSE) score and feedback-related negativity (FRN) amplitude.

**Table 1 brainsci-13-00203-t001:** Demographic and behavioral data in the MCI and HC groups.

	MCI	HC	*p* Value
Age (years)	76.2 ± 2.4	74.1 ± 3.4	*n.s.*
Sex (male/female)	6/7	6/7	*n.s.*
MMSE	25.3 ± 1.0	28.8 ± 1.3	<0.001
FAB	14.2 ± 1.2	16.0 ± 1.1	<0.001
SDS	34.8 ± 9.5	32.4 ± 6.6	*n.s.*
AS	15.6 ± 6.9	9.8 ± 5.1	0.021
RT (ms)	1289 ± 366	1060 ± 688	*n.s.*

MCI: amnestic mild cognitive impairment, HC: healthy control, MMSE: mini-mental state examination, FAB: frontal assessment battery, SDS: self-rating depression scale, AS: apathy scale, RT: reaction time, *n.s.*: not significant.

**Table 2 brainsci-13-00203-t002:** Correlation coefficients between neuropsychological data and feedback-related negativity (FRN) amplitude.

		FRN Peak Amplitude	
	Fz	Cz	Pz
MMSE	0.41 *	0.28	0.27
FAB	0.26	0.23	0.34
SDS	−0.03	−0.17	−0.06
AS	0.11	−0.12	−0.39

MMSE: mini-mental state examination, FAB: frontal assessment battery, SDS: self-rating depression scale, AS: apathy scale. * Uncorrected *p* < 0.05. Each value indicates Spearman’s correlation coefficient with adjustment for age.

## Data Availability

The data that support the findings of this study are not publicly available due to the ethical restriction but are available from the corresponding author S.A. upon reasonable request.

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
