# Peer review of "Altered Feedback-Related Negativity in Mild Cognitive Impairment"

_brainsci, 2023, doi:10.3390/brainsci13020203_

Round 1
Reviewer 1 Report
The article is devoted to studying the manifestation of FRN changes in patients with MCI during a game task. The study's relevance is justified by the fact that feedback-related negativity (FRN) is an electrical activity of the brain associated with monitoring behavior and its results. FRN is generated by negative feedback input, such as punishment or loss of money, and its potential is maximally distributed over the front central part of the skull. The authors' previous study showed that FRN latency was delayed and amplitude increased in patients with mild Alzheimer's disease (AD). Because mild cognitive impairment (MCI) is considered the prodromal stage of AD, the authors hypothesized that FRN would also be altered in MCI, as it would be in AD. Therefore, this study examined whether patients with MCI exhibited FRN changes during a play task. For this, 13 patients with MCI and 13 healthy older adults of the same age participated in a simple game task and underwent a neuropsychological assessment. Participants were asked to choose 1 of 2 options and were randomly given positive or negative feedback on their answers. An EEG was recorded during the task, and FRN was obtained by subtracting positive feedback from activity associated with negative feedback. The response time to probing stimuli was comparable in the 2 groups. Group comparison showed that FRN amplitude was significantly greater in the MCI group than in healthy older adults, but there was no group difference in FRN latency. There were no significant differences in P3 amplitude and latency between the 2 groups. The finding of increased FRN amplitude in MCI was consistent with the previous finding in AD.
Despite the satisfactory quality of the article, some shortcomings need to be corrected.
- It is recommended to expand the abstract with numerical results obtained within the study.
- The aim of the paper should be defined.
- It is recommended to include the Current research analysis section.
- The authors used data from 13 patients. It should be justified that such quantity shows reasonable results.
- It is not clear which pathologies have the patients with MCI.
- Based on the results and discussion, the Conclusion section should be included.
- The contribution of the study to the field should be defined.
In summarizing my comments, I recommend that the manuscript is accepted after major revision.
Reviewer 2 Report
This study compared the feedback-related negativity between MCI patients and healthy older adults, building on the authors’ previous research that investigated patients with Alzheimer’s disease. The results showed that FRN amplitude was greater in MCI patients compared to healthy older adults, consistent with findings in AD patients. On the other hand, FPN peak latency did not differ between the two groups. While this finding is interesting, some minor points of concern remain.
1. The authors described the result of P3 amplitude and latency in the abstract, but not in the main text. Please integrate this content by either deleting the relevant sentences in the abstract or adding them to the main text. If the latter is chosen, the authors should also discuss the results of P3 or the difference in the responsiveness of FRN and P3.
I question the adequacy of the sample size employed in this study. The Authors should consider conducting a power analysis or post-hoc sensitivity analysis to show the validity of the sample size.
3. The authors conducted partial correlation analysis for FRN and neuropsychological data with adjustment for age, but were these parametric analyses? Did the authors confirm that neuropsychological data were normally distributed? If not, the authors should use a non-parametric analysis instead.
4. It would be helpful to provide detail about the statistical values of the correlation analyses between the FRN and other neuropsychological data.
5. The discussion section does not include a description of the results of the correlation analysis between FRN and MMSE (or other parameters). The authors should discuss these results so that readers can better interpret them.
6. In lines 241-243, the authors described “Our previous study demonstrated prolonged latency of FRN in patients with mild AD, indicating that a delay of feedback information processing was associated with neural degeneration, along with the progress of AD”. But, the difference was seen with AD patients compared to healthy younger and not with healthy older, wasn't that? I think that the result of the latency of FRN in this study was similar to the authors' previous study, and the discussion in this paragraph seems a bit incorrect.
Round 2
Reviewer 1 Report
Thanks for the authors for considering reviewer's comments and recommendations. In my opinion, now the paper can be accepted.
Reviewer 2 Report
Thank you for your comment and for modifying the manuscript. The manuscript has been sufficiently amended and I no longer perceive any issues with it. I believe that it would be deemed acceptable for publication.